# Malnutrition among the aged population in Africa: A systematic review, meta-analysis, and meta-regression of studies over the past 20 years

**Temesgen Muche Ewunie**[1]*, **Habtamu Endashaw Hareru**[2], **Tadesse Mamo Dejene**[3], **Semagn Mekonen Abate**[4]

**1** Department of Human Nutrition, College of Medicine and Health Science, Dilla University, Dilla, Ethiopia,
**2** School of Public Health, College of Medicine and Health Science, Dilla University, Dilla, Ethiopia,
**3** Department of Public Health, Asrat Woldeyes Health Science Campus, Debre Birhan University, Debre Birhan, Ethiopia, **4** Department of Anesthesiology, College of Medicine and Health Science, Dilla University, Dilla, Ethiopia

* temesgenm@du.edu.et

**Data Availability Statement:** All relevant data are within the article and its Supporting Information files.

## Abstract

### Background

Nowadays, malnutrition among the advanced age (60 years and older) population is becoming a public health problem worldwide, especially in low-income countries including Africa. Hence, the prevalence in Africa is still not known. So, this review aimed to assess the pooled prevalence of under-nutrition among the advanced age population in Africa.

### Methods

A study search was carried out using databases (such as African Journals Online, Web of Science, Global Index Medicus, Embess, and PubMed) and gray literature following PRISMA guidelines from April 20, 2022, to May 30, 2022, with no restriction on date of publication. We used a standardized extraction format to compile eligible studies as per the inclusion criteria. Then, systematic review and meta-analysis were employed using a random effect model to obtain the pooled prevalence of malnutrition among aged population living in Africa. The counter-funnel plot and at the 5% significance level, Egger's test and Begg's test were used to check for publication bias. Furthermore, a meta-regression analysis was carried out to identify the relationship between the outcome of interest and different predictors.

### Results

A total of 731 studies were identified and 28 met the inclusion criteria, which were conducted in 17 African countries. The pooled prevalence of under-nutrition in Africa was 17% (95%CI; 13.5–20.6). The prevalence of malnutrition among the elderly varied significantly across countries, ranging from 1.8% (95% CI; 0.96–2.63) in South Africa to 39.47% (95% CI; 31.70–47.24) in Kenya. According to meta-regression analysis, the likelihood of a malnutrition problem would be reduced by a factor of 9.84 (β = -9.84, 95 percent CI; _-14.97, -4.70,

**Funding:** The author(s) received no specific funding for this work.

**Competing interests:** The authors have declared that no competing interests exist.

P = 0.00) in upper-middle income countries. In addition, based on the publication year, malnutrition has decreased by a factor of 0.75 (β = -0.75, 95%CI:-1.49, -0.01, P = 0.04) from 1998 to 2021.

## Conclusion

There is a high prevalence of malnutrition among the aged population. So, this underserved population should be targeted for intervention programs and/or integrated into maternal and child nutrition programs.

## Introduction

The number of geriatric population aged 60 and over is dramatically rising globally, and by 2050, it is expected to double, from an estimated 1 billion (12%) in 2020 to 2 billion (22%) [1, 2]. Increasing longevity is now creating a new challenge and these segments of the population are more vulnerable to malnutrition [3], because aging may come with cognitive and physical decline, depressive symptoms, and emotional variations [4]. These factors collectively increase the prevalence of malnutrition among elders [5]. Therefore, malnutrition among the geriatric population is a public health problem in low and middle-income countries [6–8] such problem is more serious in developing countries because of poverty, low dietary diversity [9], and comorbidity [10]. Furthermore, evidence suggests that malnutrition is more prevalent in the geriatric population, but little attention has been given [11].

Malnutrition among elders is becoming one of the major public health concerns in developing countries that cause high numbers of mortality [12]. The problem is underestimated and persistent in low-income countries, including the continent of Africa [13].

Despite numerous studies that have been conducted on malnutrition, data on the prevalence of malnutrition among the geriatric population is limited in the African continent that could be used for nutritional intervention on such segment of the population. Therefore, this systematic review and meta-regression aimed to assess the prevalence of malnutrition among the geriatric population in Africa. Hence, evidence generated from this review will be used by nutrition program implementers, policymakers, stakeholders, and health experts to achieve sustainable development goals.

## Methods and materials

### Search strategies and selection process

We followed PRISMA guidelines [14] to search articles using different electronic databases like Web of Science, African journals online, Global Index Medicus, Embess, and PubMed without date restriction from April 20, 2022, to May 30, 2022. The search process included the following key terms like:-"under-nutrition", "malnutrition", "nutritional status", geriatric", "elders", "older", "advanced age population" along with the names of each African nation. In order to not to miss studies, we searched references of relevant articles. Following the title and abstract screening process, studies that met the eligibility criteria underwent full text review, and EndNote X9 software was used to maintain citations and manage duplicate articles in the review process.

### Eligibility criteria

Studies conducted in Africa and reported the prevalence of under-nutrition, used measurements like body mass index (BMI) and a mini-nutritional assessment tool (MNA),

observational study, conducted on the aged population (elderly), and published in English were included. But, we excluded studies that did not contain sufficient data on prevalence, conducted in countries other than African nations. In addition, studies with incomplete data and not accessible, published other than English language were not included in the analysis.

## Outcome measurements

Studies that used either BMI and/or MNA measurement to assess malnutrition in an aged population were included. Nonetheless, in the studies that used both MNA and BMI measurements, the MNA measurement result was used for this review. Thus, malnutrition among the geriatric population is the outcome of interest which was dichotomized in to malnourished or not which was assessed by either BMI or MNA. Based on 1),BMI;- Those who had BMI less than 18.5 kg/m were considered as malnourished and those whose BMI was 18.5 Kg/m and above were not malnourished [15], 2),MNA:- the MNA scale was the other assessment methods that consisted of 4 nutritional areas: anthropometric measurement, dietary questionnaire, global assessment and subjective assessment. Then the sum of the score of MNA categorized into malnourished (if the score < 17), and not malnourished (if the score 17 and above) [16].

## Data extraction

Identification of eligible studies were done using Microsoft excel sheet by three independent researchers (TME, TMD, and HEH) and disagreement was resolved in all process by discussion with the fourth researcher (SMA). For each eligible study, the following information was extracted: the name of the author(s), the year of publication, the study country, the outcome measurement (BMI, MNA), the study design, the response rate, the sample size, and the prevalence of malnutrition with a 95% confidence interval. In addition, the countries' income economy level, based on the recent World Bank economic classification [17], was included in the data extraction process.

## Study quality assessment

The quality of studies was assessed by two reviewers (TME and HEH) rigorously using the Newcastle–Ottawa scale for cross-sectional studies [18]. The quality tool consists of a total of 10 questions assessing different aspects of the study, and each question is scored as yes (1), no (0), or not applicable (N/A). Finally, those articles that scored six or above out of a total of 10 criteria were included in this review.

## Statistical analysis

The outcome was the proportion of malnourished elders and was dichotomized into malnourished and not malnourished. The results were presented as a percentage with a 95% confidence interval (CI), and the analysis was carried out in STATA version 16 software using the metan function in the meta-package. The $I^2$ statistic was checked to determine the studies' heterogeneity, which describes the percentage of total variation among studies that was due to heterogeneity rather than chance. When $I^2$ exceeds 75%, homogeneity was considered [19]. A random-effect model was used and subgroup analysis was conducted to manage heterogeneity among studies using measurements, and the country's economic status. Sensitivity analysis was performed to determine the effect of a single study on the pooled estimate of outcome. Potential publication bias was checked by a funnel plot through observational assessment. In addition, the bias was checked by Egger's test and Begg's test at a 5% significance level. Then, a non-parametric trim and fill analysis was performed to manage the publication bias [20]. A

meta-regression was performed to measure the dependency of outcome on the predictors and to investigate potential effect modifiers that explain any heterogeneity effect among studies. As an assumption of meta-regression analysis, a minimum of ten studies required to perform meta-regression and to see linear-correlation of outcome variable and selected predictors.

## Results

A total of 731 articles were identified, of which, 179 duplicate articles were removed by end-note, and 331 studies were excluded by assessing their title and abstract. Finally, a total of 28 met the inclusion criteria, which were from 17 African countries (Fig 1). Furthermore, from the total of 28 eligible studies, 11 studies [21–31] used MNA tool, whereas the remaining 17 studies [13, 32–47] used BMI to assess undernutrtion in aged population (Table 1).

### Prevalence of malnutrition in Africa among aged population

The pooled prevalence of under-nutrition in Africa is 17% (95%CI; 13.5–20.6) with a significant statistical heterogeneity ($I^2 = 98$, P = 0.00) (Fig 2). Substantial variation of prevalence of under nutrition among advanced age population was observed among countries ranging from 1.8% (95%CI; 0.96–2.63) in South Africa to 39.47% (95%CI; 31.70–47.24) in Kenya (S1 Fig).

The sensitivity analysis showed that there were no studies that affect the pooled estimate of malnutrition (S2 Fig). Visual assessment of publication bias was performed using the funnel plot, which revealed asymmetric distribution among studies. In addition, Begg's test (Pr > |z| = 0.010) and Eggers' test (P = 0.00) were also carried out that showed the presence of publication bias. Then meta-trim and fill analysis was performed, which quantified the effect of missed studies (Fig 3).

### Subgroup analysis

Subgroup analysis was employed by stratifying using the measurements, study setting, and income levels of the country. The prevalence of malnutrition in aged population was higher in community-based study which was 19.13% (95%CI;- 14.68, 23.59). In addition, it was found to be 19.49% (95%CI;-14.14, 24.84) by using body mass index and 23.56% (95%CI; - 19.68, 27.44) among countries with low-income economies (Table 2).

### Meta-regression

A meta-regression was also conducted to identify the relationship between the prevalence of under-nutrition among the advanced age population in Africa and different predictors such as study period, sample size, measurement, and countries' income level. The meta-regression revealed that the publication year was statically significant, meaning that the prevalence of under-nutrition was steadily decreasing through time (P = 0.04). According to the World Bank's classification of African countries' income levels [17], the prevalence of malnutrition among the geriatric population was higher in low-income economies than in lower-middle and upper-middle income economies (P>|z| = 0.00). In addition, malnutrition in Africa among aged population was significantly decreased over the last twenty years (P>|z| = 0.04). However, the relationship between malnutrition with sample size, and measurement was not statistically significant (Table 3, Fig 4).

## Discussion

In this systematic review and meta-regression, we found that malnutrition is a public health problem among aged population in Africa, and the pooled prevalence is estimated to be

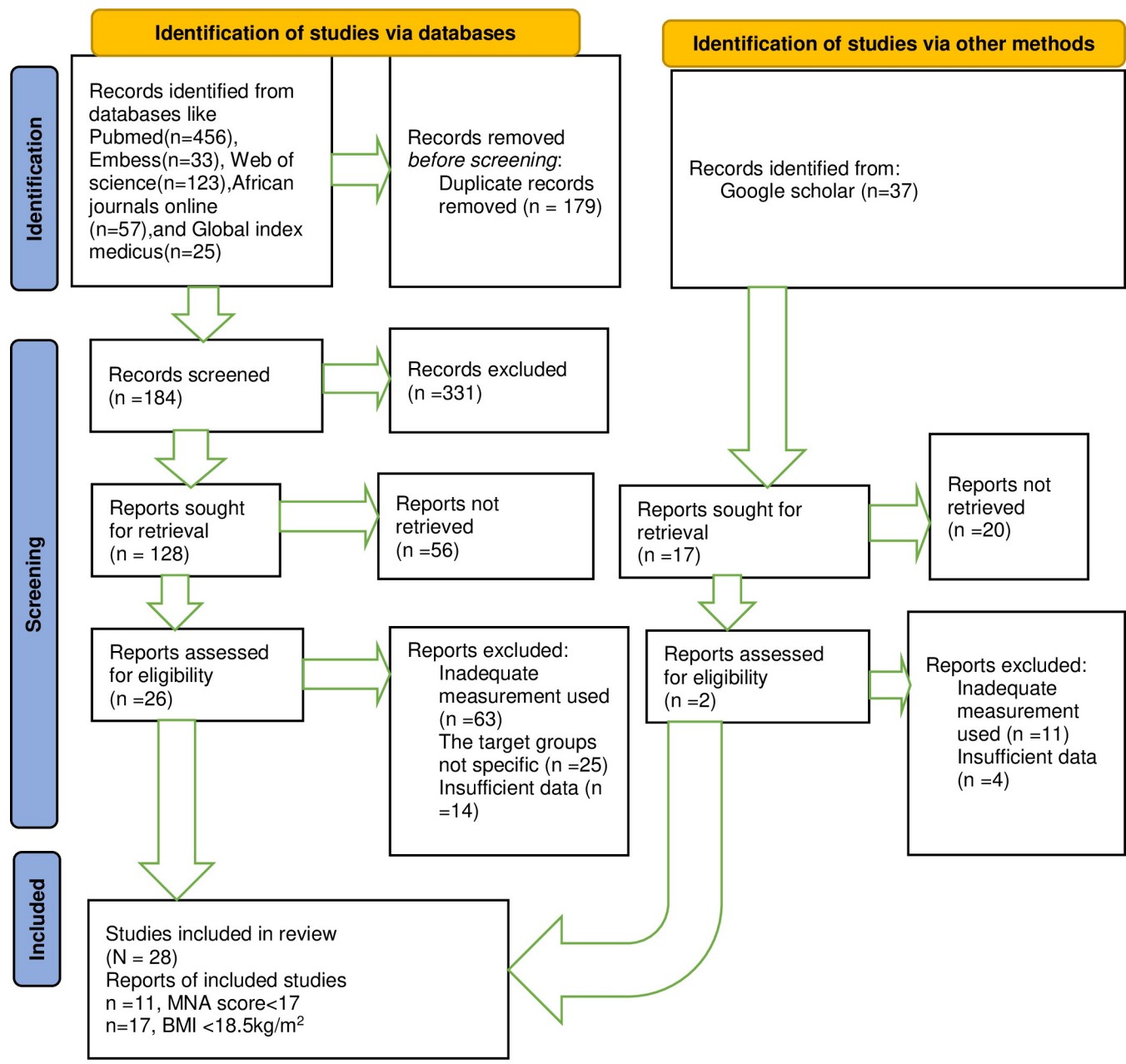

**Fig 1. A PRISMA flow diagram for systematic reviews, met analysis and meta-regression of included studies.**

17.03%. This finding is higher than the studies conducted in China (3.2%) [48], India (9.1%) [49], developed countries (3.1%) [50]. This discrepancy could be because elders who are living in Africa are in a state of food insecurity and are neglected by the healthcare system as well as limited facilities for elders. However, it was lower than the study conducted on free-living elders in Europe (53%) [51]. This probably due to the fact that those advanced age group might get family support and lower prevalence of non-communicable diseases than free-living people in Europe. This review showed that one in five geriatric people had a low body mass index ($<18.5$ mg/m$^2$), and three out of twenty geriatric people had a low MNA ($<17$ score) living in Africa. The prevalence of malnutrition varies from country to country, which ranges

**Table 1. Summary of the characteristics of included studies in the review.**

| S. No | Authors | Year of Publication | Country | Measurements | Study setting | Country's economy level | Sample size | Prevalence of malnutrition | Quality score |
|---|---|---|---|---|---|---|---|---|---|
| 1 | Pierre J et.al [35] | 2017 | Central Africa | BMI | Community-based | Low-income economy | 990 | 19.2 | 7 |
| 2 | MB Andre et.al [46] | 2013 | Republic of Congo | BMI | Community-based | Low-income economy | 370 | 34.3 | 8 |
| 3 | Aganiba BA et.al, [39] | 2015 | Ghana | BMI | Community-based | Lower-middle income | 400 | 18 | 7 |
| 4 | Geofrey M. et.al [36] | 2021 | Zambia | BMI | Community-based | Lower-middle income | 135 | 30.4 | 7 |
| 5 | DM Chilima and SJ Ismail [37] | 1998 | Malawi | BMI | Community-based | Low-income economy | 296 | 30.06 | 8 |
| 6 | Joyce K. and Fred B [42] | 2004 | Uganda | BMI | Community-based | Low-income economy | 100 | 33.3 | 6 |
| 7 | Naidoo et.al [41] | 2015 | South Africa | BMI | Community-based | Upper-middle income | 984 | 1.8 | 6 |
| 8 | W.A.O. Afolabi.et.al [47] | 2015 | Nigeria | BMI | Community-based | Lower-middle income | 140 | 2.9 | 9 |
| 9 | Dawit T et.al [33] | 2014 | Ethiopia | BMI | Community-based | Low-income economy | 757 | 21.9 | 7 |
| 10 | Kidest W et.al [43] | 2019 | Ethiopia | BMI | Community-based | Low-income economy | 554 | 17.1 | 9 |
| 11 | Gustave. M et.al [34] | 2021 | Cameroon | BMI | Community-based | Lower-middle income | 599 | 19.7 | 7 |
| 12 | Faith K et.al [45] | 2010 | Kenya | BMI | Community-based | Low-income economy | 152 | 39.47 | 8 |
| 13 | Legesse M. et.al [13] | 2019 | Ethiopia | BMI | Community-based | Low-income economy | 892 | 17.1 | 7 |
| 14 | Agbozo et.al [38] | 2018 | Ghana | BMI | Community-based | Lower-middle income | 120 | 10 | 7 |
| 15 | N. Menadi et.al [44] | 2013 | Algeria | BMI | Facility-based | Lower-middle income | 314 | 14.01 | 8 |
| 16 | Abdu O et.al [25] | 2020 | Ethiopia | MNA | Community-based | Low-income economy | 592 | 15.5 | 9 |
| 17 | Abate T et.al [30] | 2020 | Ethiopia | MNA | Community-based | Low-income economy | 662 | 26.6 | 9 |
| 18 | A. Talhaoui et.al [21] | 2019 | Morocco | MNA | Facility-based | Lower-middle income | 273 | 5.2 | 7 |
| 19 | A.Y. Abdelwahed et.al [27] | 2018 | Egypt | MNA | Community-based | Lower-middle income | 100 | 16 | 7 |
| 20 | Cheserek MJ et.al [40] | 2012 | East Africa | BMI | Community-based | Low-income economy | 573 | 26.4 | 7 |
| 21 | Andia A et.al [23] | 2019 | Niger | MNA | Community based | Low-income economy | 384 | 7.8 | 9 |
| 22 | Z.K. Adhana et.al [31] | 2019 | Ethiopia | MNA | Community-based | Low-income economy | 423 | 22.7 | 6 |
| 23 | Adebusoye LA, et.al [26] | 2011 | Nigeria | MNA | Facility-based | Lower-middle income | 500 | 7.8 | 7 |
| 24 | Hamza SA. et.al [29] | 2018 | Egypt | MNA | Community based | Low-income economy | 170 | 26.5 | 7 |
| 25 | S.A.Fattah Badr et.al [28] | 2019 | Libya | MNA | Facility-based | Upper-middle income | 312 | 11.5 | 9 |
| 26 | E.M. Mahfouz et.al [22] | 2013 | Egypt | MNA | Community-based | Lower-middle income | 350 | 8.6 | 8 |

*(Continued)*

**Table 1.** (Continued)

| S. No | Authors | Year of Publication | Country | Measurements | Study setting | Country's economy level | Sample size | Prevalence of malnutrition | Quality score |
|---|---|---|---|---|---|---|---|---|---|
| 27 | Nzeagwu O.C &Ozougwu, C.B [32] | 2019 | Nigeria | BMI | Community based | Lower-middle income | 238 | 1.7 | 8 |
| 28 | Adebusoye LA et.al [24] | 2018 | Nigeria | MNA | Facility-based | Lower-middle income | 624 | 2.24 | 6 |

from 1.8 percent in South Africa to 39.47 percent in Kenya. This could be due to socio-economic and healthcare system differences between South Africa and Kenya.

In addition, the prevalence of malnutrition among advanced age group population living in Africa is higher in low income countries as compared with upper-middle income countries. This finding is supported by the study conducted in developed countries [52]. This might be because countries with low economic status could influence people's food choices, increase the burden of food insecurity, and deteriorate nutritional status of people in those countries. Along with this, those countries with low economic status could not have enough capital to prevent malnutrition among their aged population. Concerning the study setting, the prevalence of malnutrition was higher among community based studies than facility based studies. This could be the encouragement of aged population care and better health services in facilities.

Although substantial improvement in the prevalence of malnutrition has been observed, this problem is still a public health issue. These could be a different factors, for instance aged

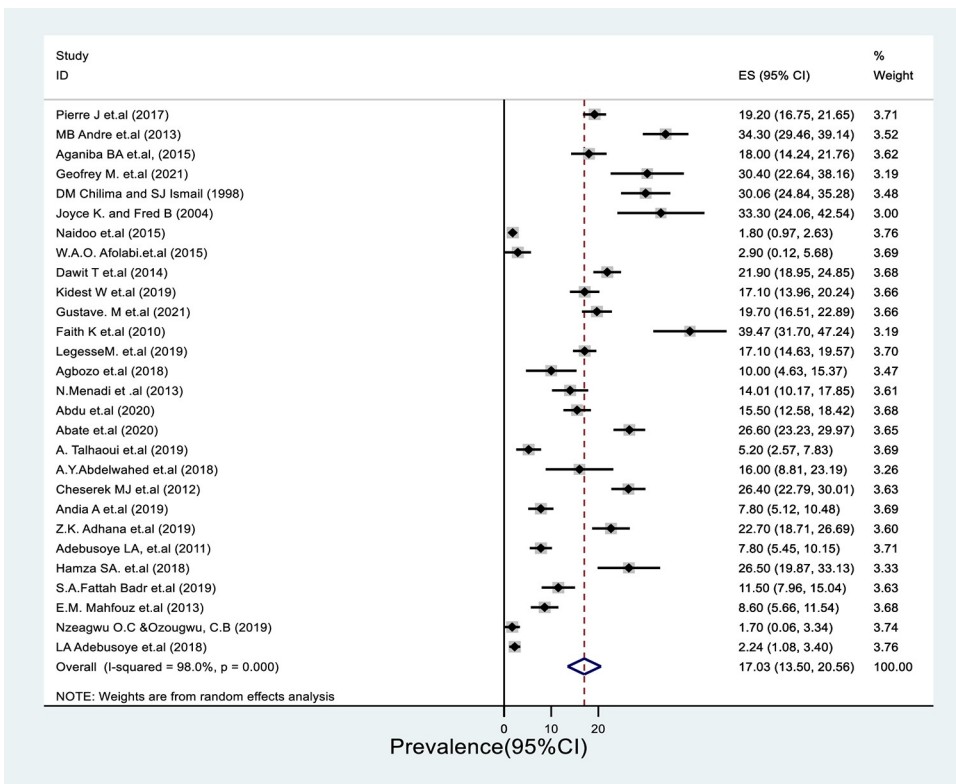

**Fig 2. Forest plot of all included studies to assess the pooled prevalence of malnutrition among aged population in Africa.**

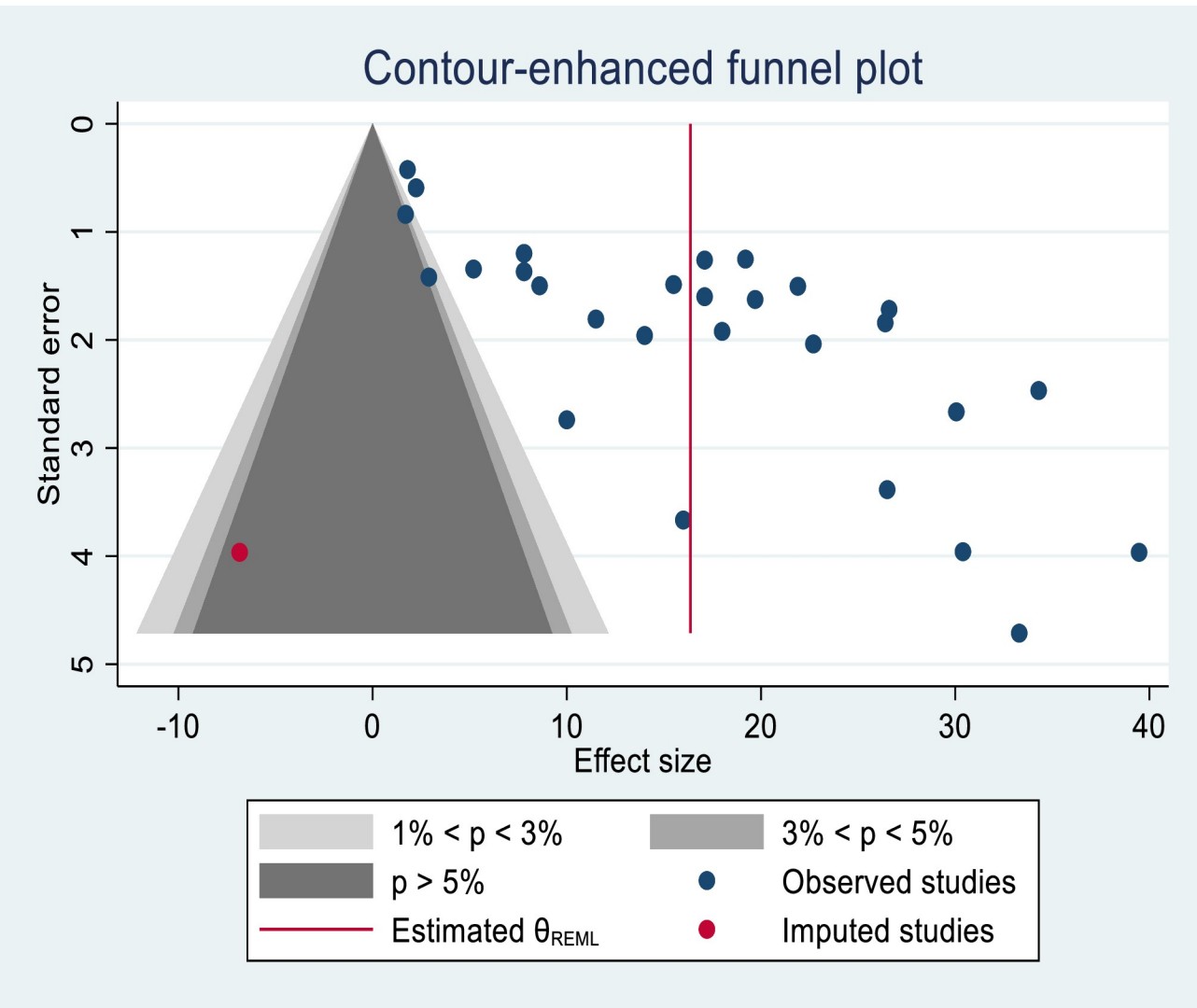

**Fig 3. Funnel plot, and trim and fill result of the analysis of included studies.**

**Table 2. Subgroup analysis of pooled prevalence of malnutrition among aged population in Africa.**

| Variables | Categories | No. of studies | Prevalence (%) | 95%CI | I-squared (%) |
|---|---|---|---|---|---|
| Measurement | BMI | 17 | 19.49 | 14.14, 24.84 | 98.4 |
| | MNA | 11 | 13.42 | 13.3, 18.27 | 97 |
| Study setting | Community-based | 23 | 19.13 | 14.68, 23.59 | 98.2 |
| | Facility-based | 5 | 7.93 | 3.73, 12.13 | 93.4 |
| Income level | Low-income economies (LIE) | 14 | 23.56 | 19.68, 27.44 | 93.9 |
| | Lower-middle-income economies (LMIE) | 12 | 10.78 | 7.06, 14.50 | 95.6 |
| | Upper-middle-income economies (UMIE) | 2 | 6.49 | -3.01,15.99 | 96.3 |
| Pooled prevalence | | 28 | 17.03 | 13.50, 20.56 | 98 |

**Table 3. Univariate meta-regression analysis result for prevalence of malnutrition among aged population in Africa.**

| S.No | Variables | P>|z| | ß-Coefficient (95%, CI) |
|---|---|---|---|
| 1 | Study period | 0.04 | -0.75 (-1.49, -0.01) |
| 2 | Sample size | 0.44 | -0.01 (-0.02, 0.01) |
| 3 | Measurements | 0.13 | -5.92 (-13.76, 1.91) |
| 4 | Income level | 0.00 | -9.84 (-14.97, -4.70) |

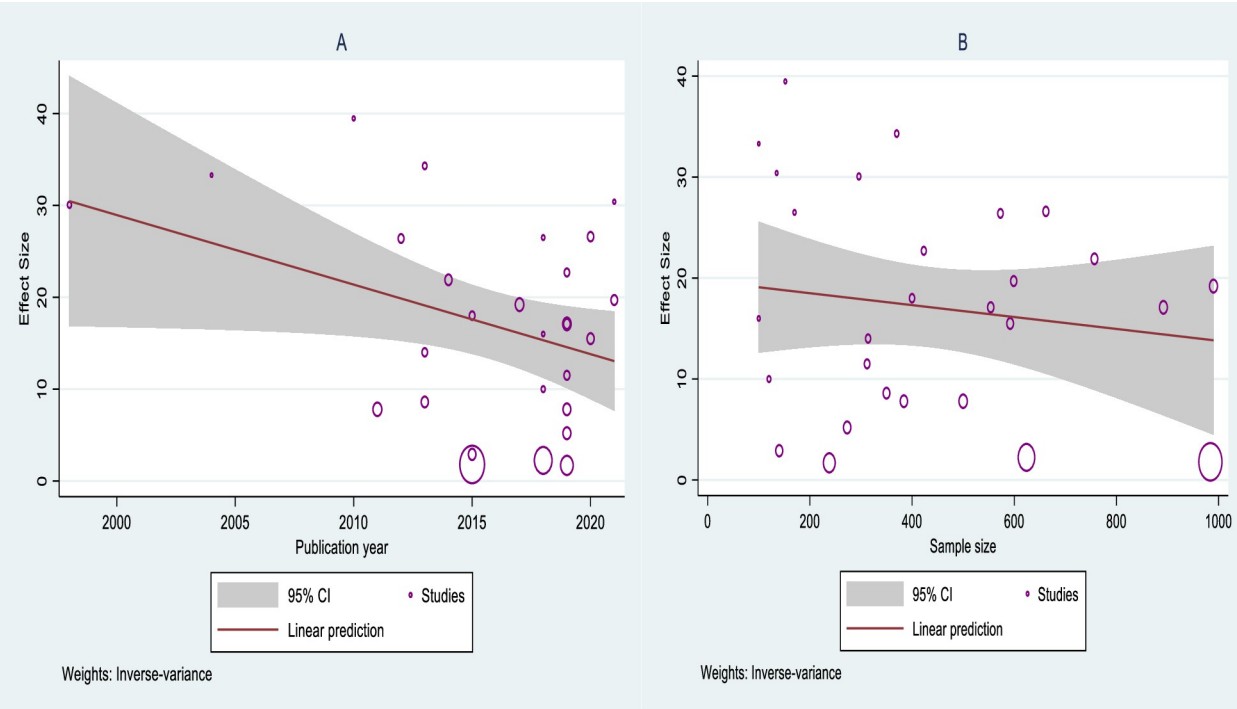

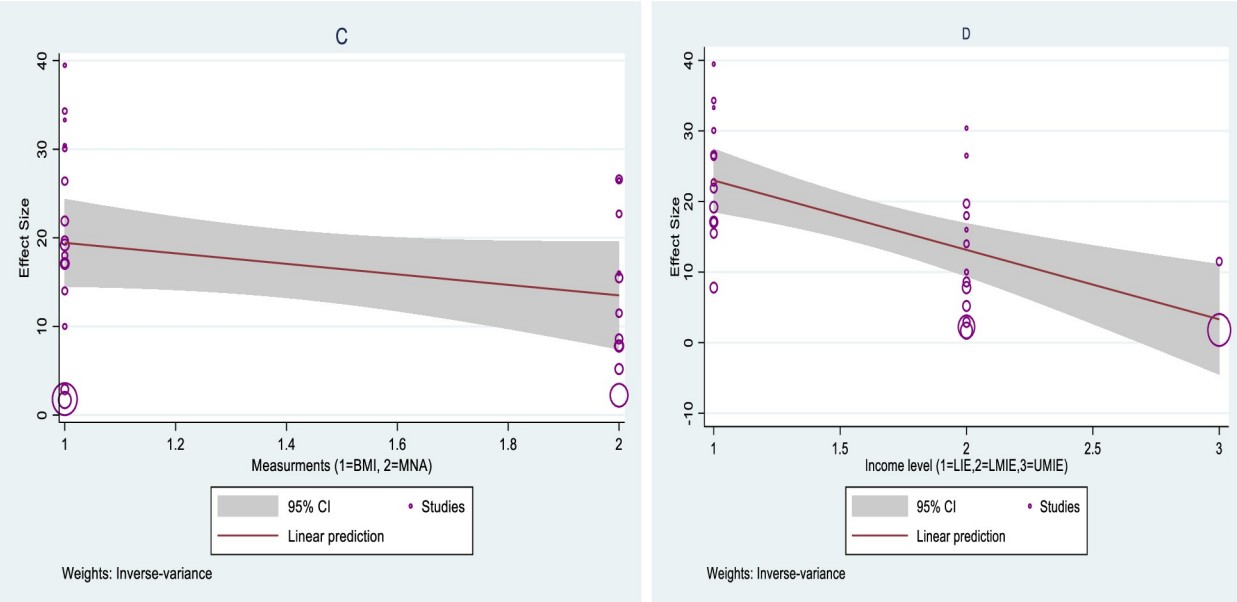

**Fig 4. Linear relationship between malnutrition and selected predictors.**

population are risky for lack of ability to chew and swallow foods, and loss of appetites that lead to malnutiriion. As a result, specific strategies to prevent malnutrition among these neglected segment of the population should be a priority to integrate with existing healthcare systems and nutrition programs.

The strength of this review could be the baseline for nutrition program implementers and policymakers to propose possible policies to integrate with the existing health care system based on the prevalence of malnutrition among advanced age group. Whereas, the limitations were determinants of malnutrition are not addressed, and articles published other than English language were not included in this review. In addition, we used of both (MNA and BMI) measurements that might be affect the estimation of prevalence of malnutrition. Future researchers should include the possible determinants. Hence, the governments of African countries should give attention to the advanced age population, which will be doubled in the near future because of the increments in life expectancy in developing countries.

Nowadays, policymakers, program implementers, and nongovernmental organizations are focusing on maternal and child nutrition. However, this review showed that there is a high prevalence of malnutrition among the aged population. As a result, this underserved population should be targeted and/or integrated into maternal and child nutrition programs, and facilities should be established for aged population care services. National and international policy arena, nutritional intervention focusing on prevention and treatment of malnutrition among the aged population should be considered.

## Supporting information

**S1 Fig. Pooled prevalence of malnutrition among African countries.**
(TIF)

**S2 Fig. Sensitivity analysis plot for the pooled prevalence of malnutrition among aged population in Africa.**
(TIF)

**S1 Table. PRISMA 2020 checklist.**
(DOCX)

## Author Contributions

**Conceptualization:** Temesgen Muche Ewunie.

**Data curation:** Temesgen Muche Ewunie, Habtamu Endashaw Hareru.

**Formal analysis:** Temesgen Muche Ewunie, Tadesse Mamo Dejene, Semagn Mekonen Abate.

**Investigation:** Temesgen Muche Ewunie, Habtamu Endashaw Hareru, Tadesse Mamo Dejene.

**Methodology:** Temesgen Muche Ewunie, Semagn Mekonen Abate.

**Software:** Temesgen Muche Ewunie, Habtamu Endashaw Hareru, Tadesse Mamo Dejene, Semagn Mekonen Abate.

**Supervision:** Semagn Mekonen Abate.

**Validation:** Temesgen Muche Ewunie, Habtamu Endashaw Hareru, Tadesse Mamo Dejene.

**Writing – original draft:** Temesgen Muche Ewunie, Habtamu Endashaw Hareru.

**Writing – review & editing:** Temesgen Muche Ewunie, Habtamu Endashaw Hareru, Tadesse Mamo Dejene, Semagn Mekonen Abate.

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
