## [Decision Letter · Decision Letter 0]

6 Sep 2022

PONE-D-22-18456Malnutrition among the aged population in Africa: A systematic review, meta-analysis, and meta-regression of studies over the past 20 years.PLOS ONE

Dear Dr. Muche,

Thank you for submitting your manuscript to PLOS ONE. After careful consideration, we feel that it has merit but does not fully meet PLOS ONE’s publication criteria as it currently stands. Therefore, we invite you to submit a revised version of the manuscript that addresses the points raised during the review process.

Please submit your revised manuscript by Oct 21 2022 11:59PM. If you will need more time than this to complete your revisions, please reply to this message or contact the journal office at plosone@plos.org. Please include the following items when submitting your revised manuscript:

We look forward to receiving your revised manuscript.

Kind regards,

Tefera Chane Mekonnen, Master in Public Health(MPH)

Academic Editor

PLOS ONE

Journal Requirements:

Additional Editor Comments:

Further clarification should be included why they pooled estimate of under nutrition among older population is a necessity from the two measurements(BMI and MNA).  Could it provide precise and valid evidence for determining malnutrition burden?The authors should consider two scenario: 1) using a  valid measurement tool(single tool) to define their primary outcome of interest, generate the findings using such indicator (e.g. MNA <17); 2)  use additional measurement modalities utilized for malnutrition screening or diagnosis and do sensitivity and subgroup analysis to see the possible heterogeneity. BMI and MNA are highly correlated since BMI is used as one component to compute MNA. Does the current estimate reflect the prevalence of undernutrition among older age individual? The setting of the study must be specified to link the current estimate with what is supposed to be( facility based or community based). The discussion section is not analytically organized, shallow and should forward the practical implication of the study and limitation needs some additional amendment. The author should mention the assumptions used in the meta-regression.The reviewer comments should be address in detail manner.

Reviewers' comments:

Reviewer's Responses to Questions

**Comments to the Author**

1. Is the manuscript technically sound, and do the data support the conclusions?

Reviewer #1: Yes

Reviewer #2: Yes

Reviewer #3: Partly

2. Has the statistical analysis been performed appropriately and rigorously? 

Reviewer #1: Yes

Reviewer #2: Yes

Reviewer #3: Yes

3. Have the authors made all data underlying the findings in their manuscript fully available?

Reviewer #1: Yes

Reviewer #2: Yes

Reviewer #3: Yes

4. Is the manuscript presented in an intelligible fashion and written in standard English?

Reviewer #1: Yes

Reviewer #2: Yes

Reviewer #3: No

5. Review Comments to the Author

Reviewer #1: Good effort. Well ompiled and systematically presented. Commendable job. The article is Technically sound. The objectives are attained. The data and information needed are all avaliable. The language is simple and lucid.

Reviewer #2: #51 Key terms: Please add meta-analysis

Eligibility criteria:

#81 what happens to the other publications not in English? Will this not result in publication bias? Some eligible publications may be in other languages other than English

Discussion

#213: The limitation should include inability to access articles/studies not written in English

Reviewer #3: The manuscript try to address a very topical issue of malnutrition among geriatric population in African countries which has been neglected over long period. However, for the manuscript to be accepted for publication the authors need to improve it by making some minor corrections. Firstly, looking at the number of countries in Africa and despite using the PRISMA flow diagram for systematic reviews, the sample size used in the study may affect the validity of the findings. The manuscript is also laden with typographical and grammatical errors. For example, in the manuscript there are some specific areas i will highlight;

Lines 55, 77-79, 84-85, 115, 132, 140-141, 155-157 and 157-158 all are not grammatically correct and need to be corrected

Lines 176-177: The statement is not complete and seems to be hanging

Lines 198-200: The statement need to be rephrased

Some of the references needs to be reviewed especially where complete citations were made. There is no need to include DOI number and website where the reference was retrieved after giving full citation already.

Figure 1: Showing the PRISMA flow diagram indicated that 28 studies were included in the review, but going through the diagram I noticed a total of 27 studies (25 reports assessed for eligibility from databases and 2 reports assessed for eligibility from other methods). Therefore, this need to be crosschecked and corrected before final submission.

6. PLOS authors have the option to publish the peer review history of their article (what does this mean?). If published, this will include your full peer review and any attached files.

Reviewer #1: No

Reviewer #2: **Yes: **Emmanuel Oladipo Babafemi

Reviewer #3: No

---

## [Author Response · Author response to Decision Letter 0]

5 Oct 2022

I revised the manuscript based on the comments provided by the reviewers and editors and the response to each specific comments addressed in the file" response to reviewers"

---

## [Decision Letter · Decision Letter 1]

28 Nov 2022

Malnutrition among the aged population in Africa: A systematic review, meta-analysis, and meta-regression of studies over the past 20 years.

PONE-D-22-18456R1

Dear Dr. Ewunie,

We’re pleased to inform you that your manuscript has been judged scientifically suitable for publication and will be formally accepted for publication once it meets all outstanding technical requirements.

Kind regards,

Tefera Chane Mekonnen, Master in Public Health(MPH)

Academic Editor

PLOS ONE

Additional Editor Comments (optional):

Reviewers' comments:

Reviewer's Responses to Questions

**Comments to the Author**

1. If the authors have adequately addressed your comments raised in a previous round of review and you feel that this manuscript is now acceptable for publication, you may indicate that here to bypass the “Comments to the Author” section, enter your conflict of interest statement in the “Confidential to Editor” section, and submit your "Accept" recommendation.

Reviewer #2: All comments have been addressed

Reviewer #3: All comments have been addressed

2. Is the manuscript technically sound, and do the data support the conclusions?

Reviewer #2: Yes

Reviewer #3: (No Response)

3. Has the statistical analysis been performed appropriately and rigorously? 

Reviewer #2: Yes

Reviewer #3: (No Response)

4. Have the authors made all data underlying the findings in their manuscript fully available?

Reviewer #2: Yes

Reviewer #3: (No Response)

5. Is the manuscript presented in an intelligible fashion and written in standard English?

Reviewer #2: Yes

Reviewer #3: (No Response)

6. Review Comments to the Author

Reviewer #2: My comments have been addressed by all the Authors as itemised on the manuscript. I therefore, recommend it for publication.

Reviewer #3: (No Response)

7. PLOS authors have the option to publish the peer review history of their article (what does this mean?). If published, this will include your full peer review and any attached files.

Reviewer #2: No

Reviewer #3: No

---

## [Editor Report · Acceptance letter]

1 Dec 2022

PONE-D-22-18456R1 

Malnutrition among the aged population in Africa: A systematic review, meta-analysis, and meta-regression of studies over the past 20 years. 

Dear Dr. Ewunie:

I'm pleased to inform you that your manuscript has been deemed suitable for publication in PLOS ONE. Congratulations! Your manuscript is now with our production department. 

Kind regards, 

on behalf of

Dr. Tefera Chane Mekonnen 

Academic Editor

PLOS ONE